# Immunogenicity of Intradermal Versus Intramuscular BNT162b2 COVID-19 Booster Vaccine in Patients with Immune-Mediated Dermatologic Diseases: A Non-Inferiority Randomized Controlled Trial

**DOI:** 10.3390/vaccines12010073

**Published:** 2024-01-11

**Authors:** Chutima Seree-aphinan, Ploysyne Rattanakaemakorn, Poonkiat Suchonwanit, Kunlawat Thadanipon, Yanisa Ratanapokasatit, Tanat Yongpisarn, Kumthorn Malathum, Pornchai Simaroj, Chavachol Setthaudom, Onchuma Lohjai, Somsak Tanrattanakorn, Kumutnart Chanprapaph

**Affiliations:** 1Department of Medicine, Division of Dermatology, Faculty of Medicine Ramathibodi Hospital, Mahidol University, Bangkok 10400, Thailand; chutima.se@psu.ac.th (C.S.-a.); ploysyne.rat@mahidol.ac.th (P.R.); poonkiat.suc@mahidol.ac.th (P.S.); yanisa.rat@mahidol.ac.th (Y.R.); tanat.yog@mahidol.ac.th (T.Y.); somsak.tan@mahidol.ac.th (S.T.); 2Department of Internal Medicine, Division of Dermatology, Faculty of Medicine, Prince of Songkla University, Songkhla 90110, Thailand; 3Department of Clinical Epidemiology and Biostatistics, Faculty of Medicine Ramathibodi Hospital, Mahidol University, Bangkok 10400, Thailand; 4Department of Medicine, Division of Infectious Diseases, Faculty of Medicine Ramathibodi Hospital, Mahidol University, Bangkok 10400, Thailand; kumthorn.mal@mahidol.ac.th; 5Department of Ophthalmology, Faculty of Medicine Ramathibodi Hospital, Mahidol University, Bangkok 10400, Thailand; pornchai.sim@mahidol.edu; 6Immunology Laboratory, Department of Pathology, Faculty of Medicine Ramathibodi Hospital, Mahidol University, Bangkok 10400, Thailand; chavachol.set@mahidol.ac.th (C.S.); onchuma.loh@mahidol.ac.th (O.L.)

**Keywords:** COVID-19 vaccines, BNT162 vaccine, intradermal vaccines, immune-mediated dermatologic disease, autoimmune bullous disease, psoriasis

## Abstract

The intradermal route has emerged as a dose-sparing alternative during the coronavirus disease 2019 (COVID-19) pandemic. Despite its efficacy in healthy populations, its immunogenicity has not been tested in immune-mediated dermatologic disease (IMDD) patients. This assessor-blinded, randomized-controlled, non-inferiority trial recruited patients with two representative IMDDs (i.e., psoriasis and autoimmune bullous diseases) to vaccinate with fractionated-dose intradermal (fID) or standard intramuscular (sIM) BNT162b2 vaccines as a fourth booster dose under block randomization stratified by age, sex, and their skin diseases. Post-vaccination SARS-CoV-2-specific IgG and interferon-γ responses measured 4 and 12 weeks post-intervention were serological surrogates used for demonstrating treatment effects. Mean differences in log-normalized outcome estimates were calculated with multivariable linear regression adjusting for their baseline values, systemic immunosuppressants used, and prior COVID-19 vaccination history. The non-inferiority margin was set for fID to retain >80% immunogenicity of sIM. With 109 participants included, 53 received fID (all entered an intention-to-treat analysis). The fID demonstrated non-inferiority to sIM in humoral (mean outcome estimates of sIM: 3.3, ΔfID-sIM [mean, 95%CI]: −0.1, −0.3 to 0.0) and cellular (mean outcome estimates of sIM: 3.2, ΔfID-sIM [mean, 95%CI]: 0.1, −0.2 to 0.3) immunogenicity outcomes. Two psoriasis patients from the fID arm (3.8%) developed injection-site Koebner’s phenomenon. Fewer fID recipients experienced post-vaccination fever (fID vs. sIM: 1.9% vs. 12.5%, *p* = 0.027). The overall incidence of disease flare-ups was low without a statistically significant difference between groups. The intradermal BNT162b2 vaccine is a viable booster option for IMDD patients troubled by post-vaccination fever; its role in mitigating the risk of flare-ups remains unclear.

## 1. Introduction

Staying up-to-date with the coronavirus disease 2019 (COVID-19) vaccination schedules is essential to protect oneself from COVID-19-related complications and maintain herd immunity to prevent another pandemic. An epidemiological model raised concerns about the waning of protective efficacy to less than 50% if the boosting schedule was lengthened to more than 1.5 years, resulting in the introduction of annual vaccination into the scientific discussion [1]. Boosting with newer vaccines may also be expected to tackle the mutating severe acute respiratory syndrome coronavirus 2 (SARS-CoV-2) variants.

Fractionated-dose intradermal vaccination (fID) is an alternative delivery route that has gained popularity among countries experiencing vaccine supply shortages as it allows up to 80% cost reduction and facilitates efficient vaccine distribution in these areas [2]. As a booster dose, fID using either mRNA or viral vector vaccines induces a less intense overall reactogenicity while offering slightly lower but acceptable protective immunity than the standard intramuscular injection (sIM) [3,4,5,6,7,8,9]. This feature of fID may help persuade more patients with immune-mediated diseases to vaccinate, as studies have shown that they hesitate to receive COVID-19 vaccines due to the possibility of vaccine-related adverse reactions that may act upon their conditions, especially in causing disease flare-ups [10]. In Thailand, where fID is offered country-wide, many patients who seek ways to mitigate the immune-triggering effect of COVID-19 vaccination often consider this vaccination option, including patients with immune-mediated dermatological diseases (IMDD).

The effects of the fID COVID-19 vaccination on IMDD patients are difficult to predict, especially when their responses to the vaccines deviate from the general population [11,12]. Previous observational studies reported a reduced COVID-19 vaccine immunogenicity in approximately half of autoimmune bullous disease (AIBD) patients following the primary series and a few after the third additional or booster dose [11,12,13,14]. Despite the preserved immune responses by the vaccines, psoriasis patients reported a higher rate of systemic adverse events, especially fever and flu-like symptoms, following immunization (AEFI) than in healthy volunteers and other IMDD patients [11,15,16]. Psoriasis is also one of the most common pre-existing conditions among people who reported cutaneous vaccine-related adverse reactions, with the incidence of psoriasis flare-up varying between studies [15,17,18,19]. Given this information, it is uncertain whether fID is an immunogenically sensible choice of booster for IMDD patients from an immunogenicity or reactogenicity perspective.

Therefore, this study aims to demonstrate non-inferiority in immunogenicity between fID and sIM in delivering the BNT162b2 COVID-19 vaccine booster dose to IMDD patients.

## 2. Materials and Methods

### 2.1. Study Design and Participants

This assessor-blinded, open-label, randomized-controlled, non-inferiority trial was conducted at the dermatology outpatient clinic in Ramathibodi Hospital, Mahidol University, Thailand. The timing of this study coincided with the period when most people in Thailand had already received the primary series and the third dose and were seeking the fourth booster dose. IMDD patients were screened for eligibility and included if they were aged ≥18 years, diagnosed with psoriasis or AIBD (e.g., pemphigus and pemphigoid groups), completed a two-dose primary series and one booster dose lasting for ≥3 months, and agreed to receive the fourth dose of the BNT162b2 COVID-19 vaccine. Patients with a history of COVID-19 infection, uncontrolled disease activity of IMDD, concomitant diagnosis of non-dermatologic immune-mediated diseases, congenital or acquired immunodeficiency syndrome, active cancer, pregnancy, and allergy to components of the BNT162b2 COVID-19 vaccine were excluded. Types of prior COVID-19 vaccines received were not restricted, as Thai people had limited choices of accessible vaccines during the pandemic. Moreover, the participants’ diagnoses, representing the two common IMDDs, were later randomized to balance their tentative immunological responses to COVID-19 vaccines extrapolated from our previous study (i.e., the immunogenicity of COVID-19 vaccines is higher among psoriasis patients compared to AIBD patients, with a reverse trend for the vaccine-related adverse reactions) [11]. The study protocol was approved by the Human Research Ethics Committee, Faculty of Medicine Ramathibodi Hospital, Mahidol University (MURA 2022/238) and prospectively registered in clinical trial databases (ClinicalTrials.gov: NCT05406908, Thai Clinical Trial Registry: TCTR20220317008). No change was made to the trial design or protocol after the trial had commenced.

### 2.2. Interventions

This study contained an investigational arm and an active comparator. The placebo and inoculation control arms were not included due to ethical concerns about the risks of disease flare-up without direct benefits. The injection site for both arms was the deltoid area of the non-dominant arm. The vaccine was reconstituted per the manufacturer’s instructions with identical appearance and viscosity between arms, leaving the difference only for the vaccine’s volume and the injection equipment. In the investigational arm, 33% fractionated-dose BNT162b2 COVID-19 vaccine (i.e., 10 µg/0.1 mL) was administered intradermally. The immunological equivalence of this dose fractionation was postulated by a modelled relationship summarized from various dose-ranging studies [20] and supported by studies in a healthy population [4,7]. The active comparator was the on-label standard dosage of BNT162b2 COVID-19 vaccine (i.e., 30 µg/0.3 mL) administered intramuscularly. We minimized variation in the inoculation effect by having a single dermatologist perform the injection and photographically document the presence of intradermal wounds for every subject in the interventional arm.

### 2.3. Randomization, Allocation, Concealment, and Blinding

Participants were allocated to either interventional arm in a 1:1 ratio using computer-generated permuted block randomization, stratifying by age (<, ≥65 years old), sex, and the disease groups (i.e., psoriasis or AIBD). Upon enrolment notification, envelopes containing the allocation sequence were delivered to the investigators, who then enrolled participants sequentially according to the number labelled, performed the vaccine injection, and covered the injection site with an opaque bandage. All participants were instructed to refrain from sharing the intervention details with the assessors. If applicable, participants were advised to discontinue methotrexate or mycophenolate mofetil for 1 week post-vaccination. They were assessed by blinded investigators for 30 min following vaccination without removing the bandage for immediate adverse reactions. The same assessors followed participants to document adverse events and disease activity of IMDD at 1-, 2-, 3-, 4-, 8-, 12-, and 24 weeks post-vaccination. Participants who developed fever or symptoms of respiratory tract infection during the study period were instructed to report the symptoms, the date of illness, and COVID-19 testing results to their assessors.

### 2.4. Study Outcomes

Serological surrogates of SARS-CoV-2-specific immunity, namely anti-SARS-CoV-2 S1 receptor binding domain IgG (referred to as anti-SARS-CoV-2 IgG from this point forward) and the interferon (IFN)-γ response induced by the SARS-CoV-2 IFN-γ release assay (IGRA), were used for immunogenicity evaluation of the interventions. The anti-N protein antibody was not evaluated in the spirit of creating a homogenized protocol for testing the efficacy of S-protein-based COVID-19 vaccines applicable to participants with mixed prior vaccination backgrounds. All tests were performed by trained laboratory personnel using automated machines. SARS-CoV-2 IgG and IFN-γ levels quantitatively represent the magnitude of humoral and cellular immune responses to the interventions, respectively. The SARS-CoV-2 IgG level was measured by the SARS-CoV-2 IgG II SEMI QUANT assay (Abbott, Chicago, IL, USA) and reported in binding antibody units per milliliter (bau/mL). The positive cut-off recommended by the manufacturer is 7.1 bau/mL. The IFN-γ response was evaluated by the Quan-T-Cell SARS-CoV-2 assay (Euroimmun, Lübeck, Germany). A detailed description of the test can be found in our previous publication [11]. The IFN-γ level above 200 milli-international units per milliliter (mIU/mL) was considered a positive response based on validation studies [21,22]. These cut-offs were used to summarize participants’ baseline humoral and cellular SARS-CoV-2-specific immunity. Outcome analyses were executed with their absolute values since the population-specific cut-off value was not available at the time of the study.

The primary outcomes were peak humoral and cellular immunogenicity observed at 4 and 12 weeks post-intervention, respectively. The secondary outcomes included SARS-CoV-2 IgG levels measured at 12 and 24 weeks post-intervention, IFN-γ responses measured at 24 weeks post-intervention, vaccine-related adverse events, post-intervention COVID-19 infection, and disease flares during the study period. Vaccine-related side effects were documented using the list modified from the World Health Organization’s (WHO) AEFI form. Documentation of disease flare episodes was performed with pre-defined criteria accompanied by objective evidence of an increased disease-specific severity score and an up-titration of treatment for disease control. Further details regarding adverse event monitoring (i.e., the list of items monitored and the operational definitions) can be found in the Appendix A (See Appendix A).

### 2.5. Assay Sensitivity Evaluation

The historical placebo-controlled trial of the fID COVID-19 vaccine is unavailable. Therefore, a relatively stable SARS-CoV-2-specific immunity level was assumed for the placebo arm since vaccine-induced immunity levels would plateau 3–6 months following the third COVID-19 vaccine, [23,24] coinciding with the time between enrolment and primary outcome measurement in this study. The effect size of the placebo was deduced from the possible variations in the serological surrogate values (i.e., the within-laboratory measurement variability of the laboratory tests), which are 4.2–6.4% for SARS-CoV-2 IgG II SEMI QUANT assay and 2.3–7.5% for Quan-T-Cell SARS-CoV-2 according to the assays’ manuals. Therefore, to ensure that the treatment effect observed is beyond that of the placebo, the goal was set for the interventions to induce a ≥10% increase in the primary outcome estimates from their baseline values to fulfill the assay sensitivity assumption.

### 2.6. Statistical Analysis

Both intention-to-treat (ITT) and per-protocol (PP) analyses were performed. During the ITT analysis, imputation using group means was conducted for patients with missing outcomes. The PP analysis of immunogenicity outcomes included participants without missing outcomes whose immunogenicity outcomes fulfilled the assay sensitivity assumption and did not acquire a breakthrough COVID-19 infection. The PP analysis of reactogenicity-related outcomes included participants without missing data. Non-inferiority analyses were performed for all immunogenicity outcomes; the outcome estimates were log-normalized before analysis. The non-inferiority margin was constructed with a synthesis method based on previous data that associated dose fractionation of BNT162b2 COVID-19 vaccine with clinical efficacy and a mean neutralizing antibody level at approximately 80% of those induced by the standard dosage (i.e., the between-arm differences in immunogenicity levels should be less than 20% of the mean levels produced by sIM) [20]. The difference in treatment effects was calculated via multivariable linear regression models, using log-transformed outcome estimates as dependent variables and their corresponding baseline values along with imbalanced baseline characteristics as covariates. Non-inferiority was interpreted using a one-sided confidence interval (CI) approach to the covariate-adjusted mean differences in log-transformed outcome estimates between the interventional arms.

Secondary reactogenicity outcomes (i.e., the percentages of participants who reported each AEFI item, were diagnosed with disease flares, and had breakthrough COVID-19) were compared between groups using chi-square or Fisher’s exact tests as appropriate and a *p*-value of 0.05 as a statistical significance threshold.

### 2.7. Sample Size Calculation

Due to the lack of prior data, serological surrogate levels measured from IMDD patients receiving the third dose as an mRNA vaccine were used for sample size estimation [13]. These levels were assumed to be similar to those measured following the fourth dose, as this trend was demonstrated in healthy volunteers [25]. To compare the difference in effect sizes under the non-inferiority criteria above with a one-sided significance level of 2.5%, a power of 80%, and an allocation ratio of 1:1, the sample sizes required to reject the null hypothesis (i.e., fID is inferior to sIM) are 110 and 118 for the primary humoral and cellular immunogenicity outcomes, respectively.

## 3. Results

### 3.1. Study Participants

Between June and August 2022, 109 IMDD patients participated in this study (92% of the desired sample size). The recruitment was stopped thereafter, given the release of the bivalent vaccine that replaced the monovalent version used in this study. Fifty-three participants were allocated to receive fID (Figure 1). All participants received treatment as allocated and were followed up in person or via telemedicine until the last visit. The number of participants entering ITT and PP analyses is illustrated in Figure 2. Baseline characteristics were mostly balanced between arms, except for the difference in the proportion of systemic immunosuppressant-free participants in the ITT sample (Table 1), those receiving interleukin 17/23 inhibitors in the PP analysis of the secondary humoral immunogenicity outcome (fID vs. sIM: 24.2% vs. 5.0%, *p* = 0.017), and participants receiving interleukin 17/23 inhibitors in the PP analysis of the cellular immunogenicity outcome (fID vs. sIM: 36.0% vs. 10.7%, *p* = 0.028). All factors showing a baseline imbalance were adjusted during multiple linear regression analyses. The number of participants who did not fulfil the assay sensitivity assumption was similar between arms without a predilection towards one particular IMDD (See Appendix A). However, these participants tended to have higher baseline SARS-CoV-2-specific serological surrogate levels, especially in cellular immunity, than those who fulfilled the assay sensitivity assumption, but the SARS-CoV2-specific immunity levels observed during the study period were comparable between the two groups.

### 3.2. Immunogenicity Outcomes

During ITT analysis, fID induced a slightly lower anti-SARS-CoV-2 IgG level than sIM at Week 4 but returned to a comparable level at Week 24 (Figure 3). In the fID arm, anti-SARS-CoV-2 IgG increased by 2053.7 (95%CI: 1163.7, 2943.6) bau/mL from baseline to peak and dropped by 186.4 (95%CI: −1652.1, 2025.0) bau/mL between Week 4 and 12 and 1034.4 (95%CI: −383.2, 2451.9) bau/mL between Week 12 and 24. At Week 4, sIM raised anti-SARS-CoV-2 IgG by 2530.6 (95%CI: 1708.6, 3352.5) bau/mL from baseline before its level decreased by 1260.7 (95%CI: 532.3, 1989.1) bau/mL between Week 4 and 12 and 768.0 (95%CI: −235.9, 1771.9) bau/mL between Week 12 and 24. After adjusting for effects of covariates (i.e., baseline SARS-CoV-2 IgG level, the use of immunosuppressants, prior COVID-19 vaccines received, and the doses of mRNA vaccines received previously), the mean difference in the log-transformed primary humoral immunogenicity outcome estimate between fID and sIM of −0.1 (95%CI: −0.3, 0.0) was shown. According to the non-inferiority margin of −0.7, the fID vaccine was non-inferior to sIM (Figure 4). ITT analyses of secondary humoral immunogenicity outcomes yielded the same conclusion as the primary one (See Appendix A). The results of PP analyses also followed those observed during ITT analyses (See Appendix A).

At Week 12, the IFN-γ response of fID recipients increased by 1002.5 (95% CI: 140.9–1864.1) mIU/mL from baseline values, while this change varies among participants receiving sIM (mean change [95% CI]: −303.1 [−1576.7, 970.4] mIU/mL). The temporal change in IFN-γ response after Week 12 differs among participants of both arms (mean change [95% CI] of fID vs. sIM: 71.1 [−637.9, 780.1] vs. 536.2 [685.2, 1757.7]). The covariate-adjusted (same covariates as the humoral immunogenicity outcomes) difference in the log-transformed primary cellular immunogenicity outcome estimates between fID and sIM was within the predefined non-inferiority margin of −0.6, with the point estimate favoring fID (mean difference [95% CI]: 0.1 [−0.2, 0.3]). The ITT analysis of secondary cellular immunogenicity outcomes and all PP analyses of cellular immunogenicity outcomes also showed the same trend (See Appendix A).

Among the covariates adjusted during regressions, only the serological surrogate values measured before the outcome of interest were positively associated with the magnitude of the immunogenicity outcomes. Conversely, the number of doses of mRNA vaccines received prior to enrolment exerts a negative effect on the increment of SARS-CoV-2-specific IFN-γ response from Week 4 to 12 in ITT analysis but not in PP analysis (See Appendix A).

### 3.3. Vaccine-Related Adverse Reactions and Breakthrough COVID-19

Among study participants, both local and systemic adverse reactions persisted for approximately one week, with only a few patients reporting symptom persistence beyond this time (Table 2). No severe adverse reactions were reported. The distinctive local reaction observed following fID was itching associated with immediate wheal formation that persisted for 1-2 weeks (Figure 5). A bullous local reaction was not found in any interventional arm. Koebner’s phenomenon was detected in two psoriasis patients in the fID group. No AIBD patient developed Koebner’s phenomenon. Other types of local reactions were observed similarly in both arms.

Regarding the systemic side effects, the most common item reported by participants was fever (Table 2); the frequency of post-vaccination fever was significantly lower among participants receiving fID compared to those receiving sIM. A total of 12 patients were diagnosed with disease flare-ups during the study period, including six AIBD patients and six psoriasis patients. The participants diagnosed with flare-ups seemed to cluster more in the sIM arm; nonetheless, the between-group difference did not reach a statistically significant level. There was an equal number of psoriasis patients with post-vaccination flare-ups between arms (3 [5.7%] participants in the fID arms, 3 [5.4%] participants in the sIM arms). However, there were more AIBD patients with flare-ups in the sIM arm compared to the fID arm (1 [1.9%] participants in the fID arm, 5 [8.9%] participants in the sIM arm). The percentages of participants needing escalated treatment but not fulfilling flare definitions were also similar between the two interventional arms.

Sixteen participants (17 infection episodes) were diagnosed with symptomatic, microbiologically confirmed breakthrough COVID-19. All episodes were mild and manageable as outpatients, with the disease resolution occurring within 1–2 weeks. No significant difference in the incidence of breakthrough COVID-19 between the interventional arms was demonstrated.

## 4. Discussion

Alternative COVID-19 vaccination routes have been actively researched during the COVID-19 pandemic with a wide range of focuses, including efficacy, safety, ease of administration, and economic benefits. Intranasal and intradermal routes were the two promising COVID-19 vaccination options that reached the later stages of studies. Although intradermal injection is widely studied, it is currently an off-label route of administration, while the intranasal vaccine has yet to be released [4,5,7,9,26,27,28,29]. The benefits of these alternative routes are often multidimensional; for example, the intranasal COVID-19 vaccine offers an IgA-mediated local protective effect and pain-free administration, though its performance in inducing systemic immunity varies and requires further studies [26]. The intradermal route offers a robust antigen-specific immunity induction with a low amount of antigen by bringing them to the proximity of dendritic cells [30]. Compared to the intramuscular route, it is more economically and logistically efficient, does not carry neurovascular injury risks, and is associated with itch rather than pain following injection, based on our previous research and a few prior studies [4,5]. Additionally, the intradermal route is theoretically more suitable for self-adjuvanted mRNA COVID-19 vaccines than the subcutaneous route, another injectable route with an elevated risk for severe local reactions and contraindicated for adjuvanted vaccines [31,32,33,34].

Immunological equivalence of intradermal to the standard intramuscular routes was demonstrated in many vaccines [35,36]; some (i.e., influenza and polio vaccines) have been proven safe and efficacious in immunocompromised hosts [37,38]. With these promising historical data and their tendency towards lower systemic adverse reactions compared to the intramuscular counterpart, the fID COVID-19 vaccines have attracted great attention from the general public [3,4,5,6,7,8,39,40]. Many forms of fractionated intradermal COVID-19 vaccines (i.e., ChAdOx1-nCoV-19, BNT162b2, and mRNA-1273) were proven adequately immunogenic in healthy populations [3,4,5,6,8,9,36,37,39,40]. However, there is a lack of data on IMDD patients. In this study, we demonstrated that the immunogenicity of the fID BNT162b2 booster vaccine was comparable to the standard intramuscular form. Interestingly, a certain number of patients did not gain measurable benefits from vaccinating with the booster dose of either route, especially in the cellular immunity aspect. The ceiling for inducing cell-mediated immunity, observed with either natural infection or vaccines [41,42,43], was limited by T cell exhaustion. This phenomenon may antedate or get accelerated after vaccination in immune-mediated disease and cancer patients due to disease- and treatment-related factors [44]. In cancer patients, Benitez Fuentes et al. observed T lymphocytes with exhaustive phenotypes as early as after the third dose. The frequency of exhausted T cells was higher among patients with poor SARS-CoV-2-specific IFN-γ responses than those with good responses [45]. The same timeframe of T cell exhaustion is speculated for IMDD patients because we observed significant numbers of participants who did not fulfil assay sensitivity in the primary cellular immunogenicity outcome in both interventional arms. Our data also suggest previous exposure to mRNA COVID-19 vaccines as a factor contributing to this inability to boost SARS-CoV-2. Because these participants had a higher baseline IFN-γ response than those who responded well to the fourth dose, pre-booster SARS-CoV-2 IGRA testing may serve to identify IMDD patients who would gain benefit from an additional vaccine over a humoral immunity booster such as long-acting monoclonal antibodies (i.e., tixagevimab and cilgavimab), but the proper cut-off IFN-γ levels would require further studies. A trend of better cellular immunogenicity among those who fulfilled assay sensitivity may also signal us to further evaluate the superiority of the intradermal to intramuscular route in inducing SARS-CoV-2-specific T-cell immunity among IMDD patients.

Regarding side effects, a significantly lower rate of post-vaccination fever was observed in participants receiving fID, even among psoriasis patients. Although the incidence of flare-ups among psoriasis participants was lower than those estimated by the Vaccine Adverse Event Reporting System Database (14.0% vs. 26.8%), the psoriasis flare-up events seem to distribute equally between arms, suggesting that fID may not help reduce flare. The risk of flare associated with mRNA vaccines and the interruption of methotrexate after vaccination may confound the effect of fID in this matter [18,46,47]. Another potential downside of the intradermal vaccination for IMDD patients is the prolonged local reaction. This protracted cutaneous stimulation may implicate the development of Koebner’s phenomenon in psoriasis patients. According to what we observed in our participants, Koebner’s phenomenon does not always lead to systemic flare-ups. The reaction may be preventable by carefully selecting a vaccine inoculation site without surrounding active skin lesions or further optimizing the vaccine’s fractionation. If the benefit of vaccination outweighs the risk, such as in the pandemic, this local side effect is manageable by topical corticosteroids and should not preclude patients from further vaccinations. Physicians should inform psoriasis patients of this information so that they can set accurate expectations for the fID BNT162b2 COVID-19 vaccine. Aside from IMDDs being studied, no new-onset immune-mediated adverse reactions (e.g., urticaria, pityriasis rosea, alopecia areata, demyelinating diseases) were observed among participants, despite the increased incidence of these autoimmune diseases after COVID-19 vaccination found in other populations [48]. The occurrence of these conditions could be masked by the use of systemic immunosuppressants before and after intervention among participants. For certain diseases, other host factors could play a role in developing these conditions post-vaccination. For example, the median age of 57 among participants in this study has placed them in the age group with a low incidence of alopecia areata (0.1 per 1000 person-year) [49]. Demyelinating diseases are also uncommon in people of Asian descent [50,51,52].

The limitations of this study are as follows: Firstly, the sample size limits the power to interpret the reactogenicity outcome analysis and does not allow rare event comparisons such as herpes zoster reactivation, with an estimated incidence of 0.20% among COVID-19 vaccinees [53]. Secondly, an asymptomatic breakthrough infection cannot be estimated since we did not check for anti-N protein antibodies. Nonetheless, all symptomatic infections were accounted for and served as clinically relevant data for the protective effect of the intervention. Thirdly, the effect of the ipsilateral or contralateral site injections relative to the previous vaccination was not part of the study protocol. Emerging evidence demonstrated that COVID-19 vaccination in the ipsilateral arm was immunogenically superior to the contralateral arm vaccination in the neutralizing activity of anti-S IgG, median spike-specific CD8 T-cell levels, and CTLA-4 expression on spike-specific CD4 T-cells [54]. Exploring this benefit for fID could be an interesting area of further study. Lastly, the lack of fID COVID-19 vaccines’ immunogenicity data in other groups of immunocompromised hosts precludes the comparison of the vaccine’s performance between patients with different immune-mediated conditions.

The strength of this study is that it is one of very few that objectively addresses the immunogenicity and reactogenicity of a COVID-19 booster dose in completely vaccinated IMDD patients, a specific group of patients with minimal information regarding their immunological response to COVID-19 vaccines as well as intradermal vaccines in general. Findings from this study are valuable in counselling IMDD patients regarding their vaccination options and establishing new knowledge essential for the next outbreak.

## 5. Conclusions

The 33% fractionated intradermal BNT162b2 COVID-19 vaccine is immunologically not inferior to its standard intramuscular form when used as a fourth booster dose in IMDD patients. Some participants showed little change in the strength of SARS-CoV-2-specific cellular immunity, suggesting the presence of a boosting ceiling. The intradermal vaccine significantly induces less post-vaccination fever, but the benefit of reducing IMDD flare-ups was not observed. A prolonged local reaction is expected from the intradermal vaccine; Koebner’s phenomenon may occur infrequently following the injection. This information is helpful in laying down the tangible benefits and risks of this vaccination option for IMDD patients.

## Figures and Tables

**Figure 1 vaccines-12-00073-f001:**
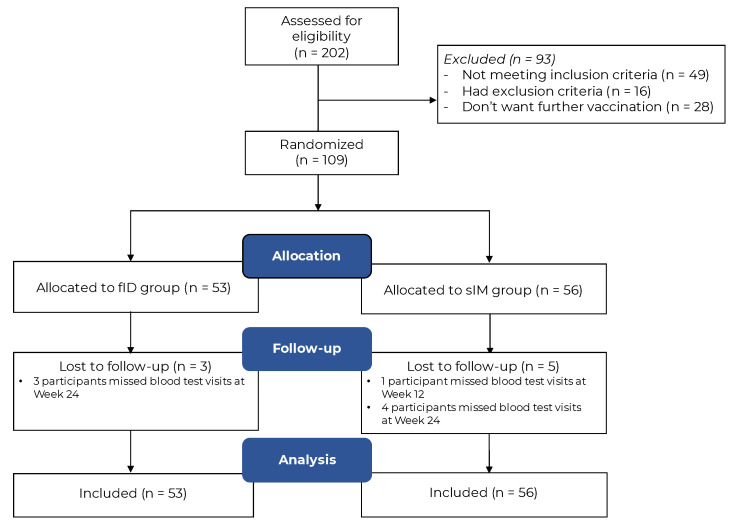
Enrollment flowchart. Patients who met the eligibility criteria and agreed to participate in the study with written informed consent were recruited. All participants received the interventions as allocated and were followed until 6 months post-intervention. Eight participants who missed some blood test visits but completed other assessments via telemedicine were included for analysis.

**Figure 2 vaccines-12-00073-f002:**
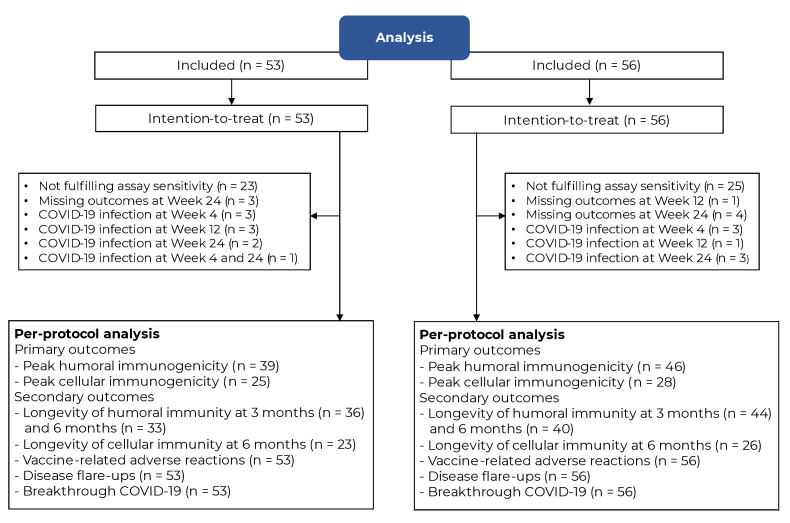
The number of participants entering intention-to-treat and per-protocol analyses. All participants entered the intention-to-treat analysis. The per-protocol analyses included participants without missing immunogenicity outcome data, fulfilled the assay sensitivity assumption, and had not contracted COVID-19 at the respective time points post-intervention. **Abbreviation:** COVID-19, coronavirus disease 2019.

**Figure 3 vaccines-12-00073-f003:**
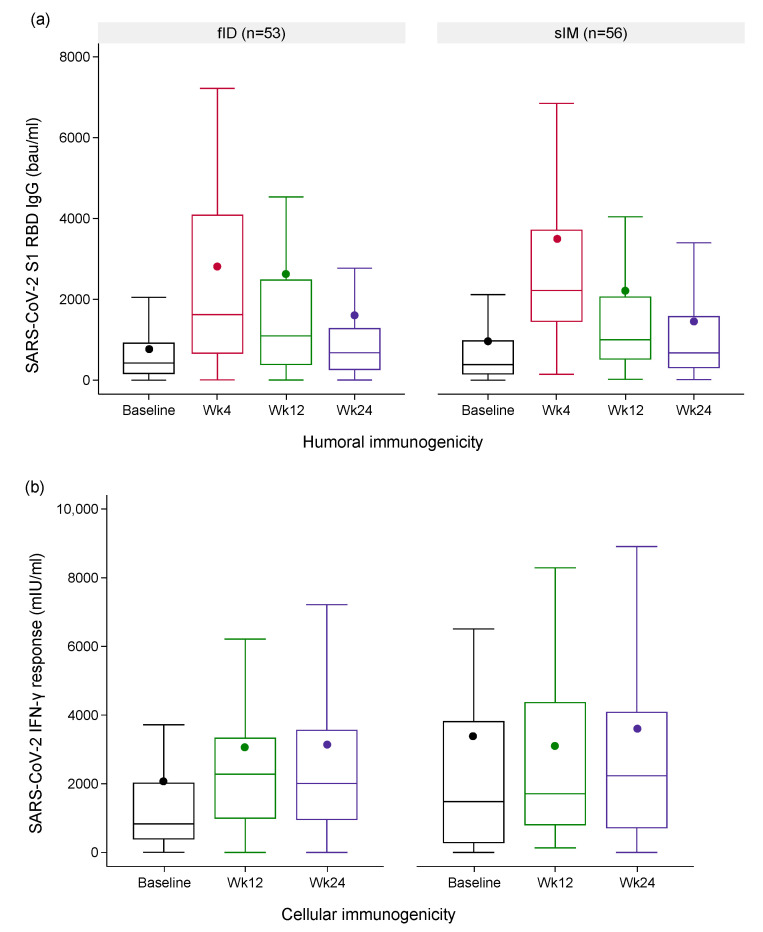
The longitudinal trend of serological surrogates of SARS-CoV-2-specific immunogenicity. The boxplots illustrate SARS-CoV-2 S1 RBD IgG levels (**a**) and IFN-γ responses measured by SARS-CoV-2 IGRA (**b**) from baseline to 24 weeks post-intervention in each interventional arm. The circles indicate the means. **Abbreviations**: bau/mL, binding antibody units per milliliter; fID, fractionated intradermal; IFN-γ, interferon gamma; IGRA, IFN-γ release assay; mIU/mL, milli-international units per milliliter; RBD, receptor binding protein; SARS-CoV-2, severe acute respiratory syndrome coronavirus 2; sIM, standard intramuscular; Wk, weeks.

**Figure 4 vaccines-12-00073-f004:**
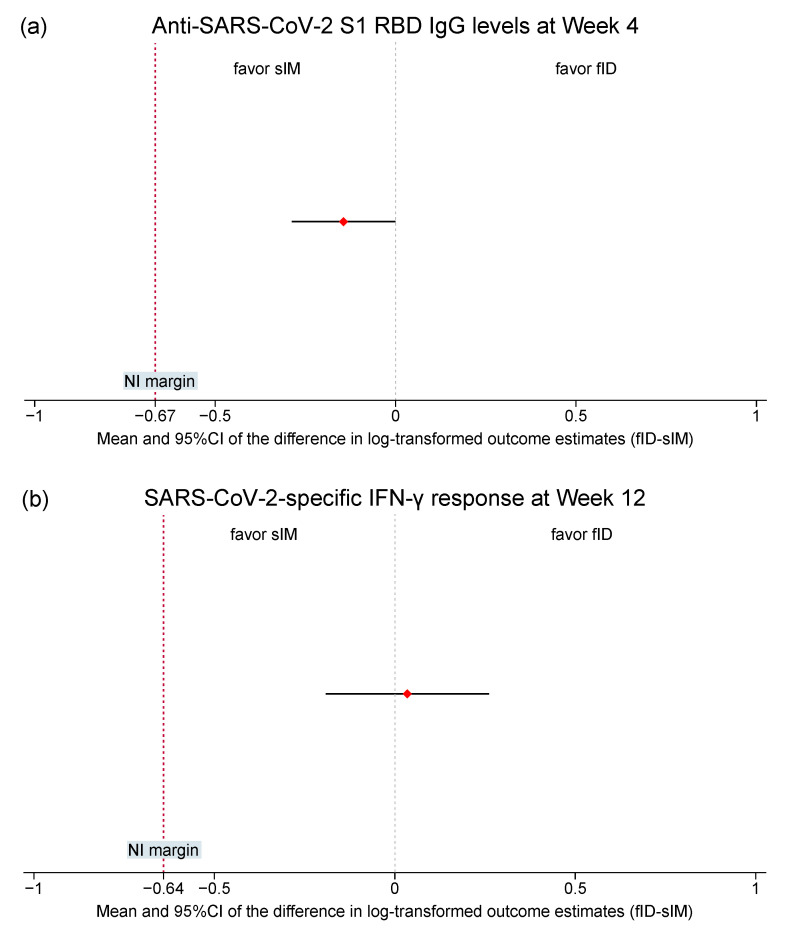
Intention-to-treat analysis of the mean difference in the primary humoral (**a**) and cellular (**b**) immunogenicity outcome estimates. Anti-SARS-CoV-2 S1 RBD IgG measured 4 weeks post-vaccination and the IFN-γ response measured by SARS-CoV-2 IGRA at 12 weeks post-vaccination were log-normalized prior to analysis. The mean outcome differences between arms were estimated using multivariable linear regression analyses, adjusting for the baseline value of the corresponding outcome measures, the use of systemic immunosuppressants, the types of COVID-19 vaccines, and the doses of mRNA vaccines previously received. Diamonds, solid black lines, and dotted red lines represent means, 95%CI, and NI margins. **Abbreviations:** CI, confidence interval; COVID-19, coronavirus disease 2019; fID, fractionated intradermal; IFN-γ, interferon gamma; IGRA, IFN-γ release assay; NI, non-inferiority; RBD, receptor binding protein; SARS-CoV-2, severe acute respiratory syndrome coronavirus 2; sIM, standard intramuscular.

**Figure 5 vaccines-12-00073-f005:**
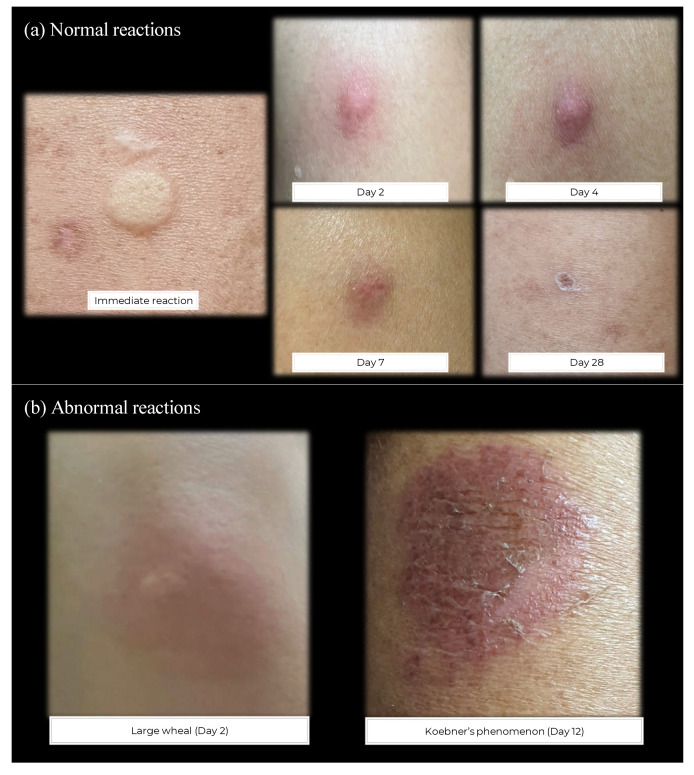
Examples of normal (**a**) and abnormal (**b**) local vaccine-related reactions following the fractionated-dose intradermal injection. All patients developed a small induration immediately after the injection, followed by a wheal formation that persisted for 1–2 weeks. The wheal gradually resolved and may leave residual post-inflammatory hyperpigmentation or a scaly erythematous patch that can be seen up to 4 weeks post-vaccination. Abnormal reactions occurred infrequently, such as a large wheal formation a few days following injection or Koebner’s phenomenon occurring around 1–2 weeks after injection.

**Table 1 vaccines-12-00073-t001:** Baseline characteristics of the study participants.

	Investigational Arms		*p*
	fID (n = 53)	sIM (n = 56)	
**Age group, n (%)**			0.700 ^a^
<65	35 (66.0)	35 (62.5)	
≥65	18 (34.0)	21 (37.5)	
**Female, n (%)**	28 (52.8)	31 (55.4)	0.791 ^a^
**Immune-mediated dermatologic diseases, n (%)**			0.887 ^a^
Autoimmune bullous diseases	31 (58.5)	32 (57.1)	
Psoriasis	22 (41.5)	24 (42.9)	
**Systemic immunosuppressants used before intervention**			
Prednisolone, n (%)	19 (35.9)	17 (30.4)	0.542 ^a^
Dose (mg/day), median (IQR)	6.3 (3.8–10.0)	5.0 (3.8–7.5)	0.501 ^b^
Azathioprine, n (%)	24 (45.3)	17 (30.4)	0.108 ^a^
Dose (mg/day), median (IQR)	62.5 (30.4–100)	50.0 (25.0–75.0)	0.257 ^b^
Methotrexate, n (%)	12 (22.6)	13 (23.2)	0.943 ^a^
Dose (mg/week), median (IQR)	10.0 (5.0–16.3)	12.5 (10.0–12.5)	0.599 ^b^
Mycophenolate mofetil, n (%)	0	3 (5.4)	0.088 ^a^
Dose (mg/day), median (IQR)	0	1000 (1000–3000)	NA
Cyclophosphamide, n (%)	1 (1.9)	0	0.302 ^a^
Dose (mg/day), median (IQR)	14.3 (14.3–14.3)	0	NA
Cyclosporin, n (%)	3 (5.7)	2 (3.6)	0.602 ^a^
Dose (mg/day), median (IQR)	50 (50–100)	125 (100–150)	0.128 ^b^
Sulfasalazine, n (%)	4 (7.6)	2 (3.6)	0.363 ^a^
Dose (mg/day), median (IQR)	2500 (1500–3000)	2500 (2000–3000)	0.803 ^b^
Leflunomide, n (%)	2 (3.8)	1 (1.8)	0.526 ^a^
Dose (mg/day), median (IQR)	20 (20–20)	20 (20–20)	1.000 ^b^
Recent rituximab use: ^c^, n (%)	13 (24.5)	10 (17.9)	0.394 ^a^
Interleukin 17/interleukin 23 inhibitors ^d^, n (%)	12 (22.6)	6 (10.7)	0.094 ^a^
Tumor necrotic factor inhibitors ^d^, n (%)	1 (1.9)	0	0.302 ^a^
No systemic immunosuppressants were used, n (%)	5 (9.4)	14 (25.0)	**0.032 ^a,^***
**Previous COVID-19 vaccination, n (%)**			
Primary series			0.487 ^a^
Viral vector vaccines	38 (71.7)	41 (73.2)	
Inactivated vaccines	9 (17.0)	9 (16.1)	
Heterologous vaccines	4 (7.6)	6 (10.7)	
mRNA vaccines	2 (3.8)	0	
Third dose			0.693 ^a^
mRNA vaccines	51 (96.2)	53 (94.6)	
Viral vector vaccines	2 (3.8)	3 (5.4)	
**Interval between the third and fourth doses (days), median (IQR)**	148 (130–178)	155 (136.0–183.5)	0.507 ^b^
**Baseline SARS-CoV-2-specific immunity levels, median (IQR)**			
Anti-SARS-CoV-2 S1 RBD IgG (bau/mL)	422.7 (153.5–927.1)	385.0 (142.2–985.1)	0.974 ^b^
% Participants tested negative (<7.1 bau/mL)	5 (9.4)	3 (5.4)	0.415 ^a^
IFN-γ measured from SARS-CoV-2 IGRA (mIU/mL)	831.3 (379.5–2031.3)	1481.4 (270.1–3822.3)	0.403 ^b^
% Participants tested negative (≤200 mIU/mL)	8 (15.1)	10 (17.9)	0.698 ^a^
**Participants whose immunogenicity data did not fulfil the assay sensitivity assumption, n (%)**			
Humoral immunogenicity outcome	1 (1.9)	1 (1.8)	0.969 ^a^
Cellular immunogenicity outcome	21 (39.6)	24 (43.6)	0.672 ^a^

** p* <0.05 **^a^** *p*-value from the chi-square or Fisher’s exact test, **^b^** *p*-value from the Mann–Whitney test, **^c^** The most recent course of rituximab treatment received by all participants was administered as follows: two doses of 1000 mg rituximab infusions separated by 2 weeks. Recent use was defined as a rituximab-to-vaccination interval < 9 months, **^d^** Biologics were prescribed with the standard dosage for psoriasis. **Abbreviations:** bau, binding antibody unit; COVID-19, coronavirus disease 2019; fID, fractionated intradermal; IFN-γ, interferon gamma; IGRA, IFN-γ release assay; IQR, interquartile range; IU, international unit; ml, milliliter; NA, not applicable; SARS-CoV-2, severe acute respiratory syndrome coronavirus 2; sIM: standard intramuscular.

**Table 2 vaccines-12-00073-t002:** Vaccine reactogenicity, disease activity, and breakthrough COVID-19.

	Investigational Arms		*p*
	fID (n = 53)	sIM (n = 56)	
**Vaccine-related local adverse reactions**			
Acute immunization site pain	40 (75.5)	41 (73.2)	0.787 ^a^
Pain score (possible range: 1–10), median (IQR)	3 (2–5)	2 (1–4)	0.493 ^b^
Delayed immunization site pain	9 (17.0)	13 (23.2)	0.418 ^a^
Pain score (possible range: 1–10), median (IQR)	3 (3–3)	5 (4–6)	0.270 ^b^
Itching, n (%)	12 (22.6)	0	<0.001 ^a,^*
Induration, n (%)	6 (11.3)	4 (7.1)	0.336 ^a^
Swelling of limb, n (%)	0	1 (1.8)	0.514 ^a^
Nodule at injection site, n (%)	0	0	NA
Abscess or cellulitis, n (%)	0	0	NA
Ipsilateral lymph node enlargement or lymphadenitis, n (%)	0	0	NA
Bleeding at injection site, n (%)	0	0	NA
Local reaction persisting for > 3 days, n (%)	3 (5.7)	2 (3.6)	0.474 ^a^
Local reaction extending beyond the nearest joint, n (%)	0	0	NA
Koebner’s phenomenon, n (%)	2 (3.8)	0	0.234 ^a^
**Vaccine-related systemic adverse reactions**			
Fever, n (%)			
No fever	52 (98.1)	49 (87.5)	
Fever < 38 °C	0	6 (10.7)	0.027 ^a,^*
Fever ≥ 38 °C	1 (1.9)	1 (1.8)	
Headache, n (%)	2 (3.8)	3 (5.4)	0.234 ^a^
Chills, n (%)	0	0	NA
Arthritis, n (%)	0	2 (3.6)	0.262 ^a^
Muscle pain, n (%)	4 (7.6)	11 (19.6)	0.067 ^a^
Fatigue or tiredness, n (%)	0	4 (7.1)	0.119 ^a^
Drowsiness, n (%)	0	2 (3.6)	0.496 ^a^
Dizziness, n (%)	1 (1.9)	2 (3.6)	0.262 ^a^
Upper respiratory symptoms, n (%)	1 (1.9)	1 (1.8)	0.738 ^a^
Others: fainting, gastrointestinal symptoms, neurological conditions, systemic cutaneous reactions, anaphylaxis, thrombocytopenia, toxic shock syndrome, sepsis, n (%)	0	0	NA
**Duration of vaccine-related adverse reactions, n (%)**			
≤1 week	21 (39.6)	24 (42.9)	0.732 ^a^
>1–2 weeks	4 (7.5)	5 (8.9)	0.535 ^a^
>2–3 weeks	0	2 (3.6)	0.262^a^
**Disease activity, n (%)**			
Participants diagnosed with flare-ups during the study period	4 (7.5)	8 (14.3)	0.261^a^
Diagnosed in less than 1 month post-intervention	0	4 (7.1)	0.066 ^a^
Diagnosed after 1 month but less than 3 months post-intervention	0	2 (3.6)	0.262 ^a^
Diagnosed after 3 months post-intervention	4 (7.5)	2 (3.6)	0.313 ^a^
Participants with dose escalation of systemic immunosuppressants despite not fulfilling flare definitions	4 (7.5)	5 (8.9)	0.793 ^a^
**Participants with breakthrough COVID-19 during the study period, n (%)**	9 (17.0)	7 (12.5)	0.509 ^a^
Diagnosed in less than 1 month post-intervention	4 (7.5)	3 (5.4)	0.641 ^a^
Diagnosed after 1 month but less than 3 months post-intervention	3 (5.7)	1 (1.8)	0.288 ^a^
Diagnosed after 3 months post-intervention	3 (5.7)	3 (5.4)	0.634 ^a^

** p* < 0.05; **^a^** *p*-value from the chi-square or Fisher’s exact test; **^b^** *p*-value from the Mann–Whitney test. **Abbreviations:** COVID-19, coronavirus disease 2019; fID, fractionated intradermal; IQR, interquartile range; NA, not applicable; sIM: standard intramuscular.

## Data Availability

The data presented in this study are available on request from the corresponding author. The data are not publicly available due to the ethical committee regulations.

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
