# Peer review of "Immunogenicity of Intradermal Versus Intramuscular BNT162b2 COVID-19 Booster Vaccine in Patients with Immune-Mediated Dermatologic Diseases: A Non-Inferiority Randomized Controlled Trial"

_vaccines, 2024, doi:10.3390/vaccines12010073_

Round 1

Reviewer 1 Report

Comments and Suggestions for Authors   Immunogenicity of intradermal versus intramuscular BNT162b2 COVID-19 booster vaccine in patients with immune-mediated dermatologic diseases: a non-inferiority randomized controlled trial. Manuscript ID: vaccines-2745726 The authors have compared the immunogenicity response in intradermal vs intramuscular vaccination of in a BNT162b2 vaccine, in a  assessor-blinded, randomized-controlled, non-inferiority trial recruited vaccinated immune-mediated dermatologic disease (IMDD) patients. Post-vaccination SARS-CoV-2-specific IgG levels and interferon-γ response were used to demonstrate treatment effects. The scientists found out that the fID is immunogenically non-inferior to sIM, associated with milder systemic side effects and infrequent Koebner’s phenomenon. Overall the manuscript is well-written nd well organized. The various sections like the results, introduction, amd conclusion are well written. There are only specific comments that needs to be addressed. Oterwise the manuscript can be accepted in the current format. Major comments: None Minor comments 1. Please center Figure 1. 2. Check the quality of figure 2. The font on Figure 4 is not clear   Comments on the Quality of English Language

Moderate editing of English language required

Author Response

We would like to express our deepest gratitude for your review and constructive comments regarding our manuscript titled “Immunogenicity of intradermal versus intramuscular BNT162b2 COVID-19 booster vaccine in patients with immune-mediated dermatologic diseases: a non-inferiority randomized controlled trial.” submitted to Vaccines. We appreciate your positive comments and kind remarks on our manuscript. Alterations have been made according to your recommendations.

- Please centre Figure 1.

  • The figure was adjusted as suggested.

- Check the quality of figure 2

  • The figure was adjusted for more clarity and improvement in quality.

- The font on Figure 4 is not clear 

  • The quality of the figure was adjusted.

Reviewer 2 Report

Comments and Suggestions for Authors

In this interesting study, two methods of vaccination was tested for their efficacy of both humoral and cellular immunity of COVID19 in patients with IMDD. The results showed , in general, comparable results and may be adapted in clinical practice. Several questions and suggestions exist:

1 Why is 1/3 dose chosen for the intradermal injection?

2. Please reference for the sentence “Previous studies have demonstrated a higher rate of reported adverse events following immunization (AEFI) in psoriasis patients than in healthy volunteers and other IMDD patients.”

3. Why is the fourth dose chosen for the study, and not the 3rd? Some study already suggested that urticaria following the 3rd COVID vaccination is more resistant.

completed a two-dose primary series and one booster dose lasting for > 86 3 months, and agreed to receive the fourth dose as BNT162b2 COVID-19 vaccine.

4. Since the immunogenicity of COVID-19 vaccines is higher among psoriasis patients compared to AIBD patients with a reverse trend showed for the vaccine-related adverse reactions).[Ref 12, A real-world prospective cohort study of 441 immunogenicity and reactogenicity of ChAdOx1-S[recombinant] among patients with immune-mediated dermatological 442 diseases. British Journal of Dermatology 2022, 188, 268-277], I would suggest to add more references and explanation of psoriasis aggravation following COVID19 vaccination in controlled studies as the potential down sides. Other than humoral and cellular immunity in psoriasis, other features of fever, myalgia, malaise will be also important to discuss.

5. Are the injection sites the same, because studies have shown that injections into the same site will result in stronger immune reactions? For systemic effects, some experts have suggested injection to different sites, but for local effect (such as in the treatment of verruca), a repeated injection is suggested. Please comment on this.

6. Due to higher local effects, should patients with active/unstable psoriasis avoid intradermal vaccination?

7 There is imbalance of no systemic immunosuppressants between groups, and the potential impact should be discussed in more depth.

8 Is higher itch only on the injection sites or it is generalized? Is it part of urticarial? What about the prevalence of uritcaria, zoster and hair loss in the two groups. These are relatively common in patients receiving COVID vaccination and should be observed in at least some patients in this cohort.

9        Is muscle pain as AE only on the injection sites or it is generalized?

10. Please lists a pro-con comparison. I think it should cover difference in pain, volume, use of adjuvant, speed of mass vaccination, and the disease to be prevented, etc.  

11 Other than polio, Covid19, and influenza, are there other examples of intradermal and intramuscular studies? In addition to intradermal and intramuscular, what about subcutaneous vaccination?

Author Response

We would like to express our deepest gratitude for your review and constructive comments regarding our manuscript titled “Immunogenicity of intradermal versus intramuscular BNT162b2 COVID-19 booster vaccine in patients with immune-mediated dermatologic diseases: a non-inferiority randomized controlled trial.” submitted to Vaccines. We are grateful for your time on handling our manuscript and sincerely appreciate the opportunity to revise. The changes made in response to your comments have strengthen our manuscript considerably. Please find our responses to your comments and suggestion below

- Why is 1/3 dose chosen for the intradermal injection?

  • Ans: The immunological equivalence of the 1/3 dose fractionation used in this study (10 mg/0.1 ml instead of 30 mcg/0.3 ml) is postulated by a modelled relationship summarised from various dose-ranging studies (1) and supported by studies in a healthy population (2-3). This was a vaccination option offered by the Ministry of Health in Thailand. We added this information to the revised manuscript (Line 118-120).

References

  1. Wiecek W, Ahuja A, Chaudhuri E, et al. Testing fractional doses of COVID-19 vaccines. Proc Natl Acad Sci U S A. Feb 22 2022;119(8)doi:10.1073/pnas.2116932119
  2. Intapiboon, P.; Seepathomnarong, P.; Ongarj, J.; Surasombatpattana, S.; Uppanisakorn, S.; Mahasirimongkol, S.; Sawaengdee, W.; Phumiamorn, S.; Sapsutthipas, S.; Sangsupawanich, P.; et al. Immunogenicity and Safety of an Intradermal BNT162b2 mRNA Vaccine Booster after Two Doses of Inactivated SARS-CoV-2 Vaccine in Healthy Population. Vaccines (Basel) 2021, 9, doi:10.3390/vaccines9121375.
  3. Temtanakitpaisan, Y.; Saengnipanthkul, S.; Sarakosol, N.; Maskasame, S.; Mongkon, S.; Buranrat, B.; Thammawat, S.; Patamatamkul, S.; Nernsai, P. Reactogenicity and immunogenicity of the intradermal administration of BNT162b2 mRNA vaccine in healthy adults who were primed with an inactivated SARS-CoV-2 vaccine. Vaccine X 2022, 12, 100242, doi:10.1016/j.jvacx.2022.100242.

- Why is the fourth dose chosen for the study, and not the 3rd?.

  • Ans: Although the timing for studying the boosting effect of the COVID-19 vaccine can start from the third dose, people in Thailand were encouraged to receive the primary series and the third dose as soon as possible to control the pandemic. Given the extremely limited vaccine options and supply during that time, allocating COVID-19 for interventional research in a small target group was not feasible in our country. The pandemic situation and vaccine supply allowed the earliest time to start the study when the patients sought the fourth dose. We explained this issue in the revised manuscript (Line 89-91)
  • Nevertheless, with this current studied population of immune-mediated dermatologic disease patients, we have learned about the patient’s response to the primary series and the third dose through observational studies (1-3). Patients with different immune-mediated dermatological diseases respond differently to primary series. The vaccines’ immunogenicity in patients with autoimmune bullous diseases (AIBD) was lower than those of the healthy population, while the outcomes observed in psoriasis patients were comparable to controls. A few of these non-seroconverters did not response to the third booster dose, either as an additional or booster dose (3). This information can be found in the introduction of the revised manuscript (Line 69-76, Page 2).

References

  1. Chanprapaph K, Seree-aphinan C, Rattanakaemakorn P, Pomsoong C, Ratanapokasatit Y, Setthaudom C, et al. A real-world prospective cohort study of immunogenicity and reactogenicity of ChAdOx1-S[recombinant] among patients with immune-mediated dermatological diseases. Br J Dermatol [Internet]. 2023 [cited 2023 Dec 17];188(2):268–77. Available from: https://academic.oup.com/bjd/article-abstract/188/2/268/6765233?redirectedFrom=fulltext
  2. Seree-aphinan C, Chanprapaph K, Rattanakaemakorn P, Setthaudom C, Suangtamai T, Pomsoong C, et al. Inactivated COVID-19 vaccine induces a low humoral immune response in a subset of dermatological patients receiving immunosuppressants. Front Med (Lausanne) [Internet]. 2021;8. Available from: http://dx.doi.org/10.3389/fmed.2021.769845
  3. Seree-aphinan C, Suchonwanit P, Rattanakaemakorn P, Pomsoong C, Ratanapokasatit Y, Setthaudom C, et al. Risk–benefit profiles associated with receiving Moderna COVID‐19 (mRNA‐1273) vaccine as an additional pre‐booster dose in immune‐mediated dermatologic disease patients with low SARS‐CoV‐2‐specific immunity following the primary series: A prospective cohort study. J Eur Acad Dermatol Venereol [Internet]. 2023;37(5). Available from: http://dx.doi.org/10.1111/jdv.18890

- Is higher itch only on the injection sites or it is generalized? Is it part of urticarial?

- What about the prevalence of urticaria. Some study already suggested that urticaria following the 3rd COVID vaccination is more resistant

  • Ans: Itch was evaluated as the part of local reactions. It was evaluated separately from urticaria which were under a “systemic cutaneous reaction category. Detailed description of the adverse event monitoring can be found in the Supplementary 1. We did not observe urticaria in the study, although this cutaneous adverse event was reported frequently in various study. We believe the use of system immunosuppressants could play role in masking this side effect among participants. We discussed the matter of vaccine-related adverse reactions in the revised manuscript (Line 448-468).

- What about the prevalence of zoster and hair loss in the two groups. These are relatively common in patients receiving COVID vaccination and should be observed in at least some patients in this cohort.

  • We did not observe herpes zoster or hair loss during the study period. Cutaneous reaction is listed as separate items and inquired in each visit. Frequent study follow-up, the ability to directly contact the assigned assessors, and their regular visits to the dermatology clinic made it hard to miss these conditions if they did come up during the study. Herpes zoster, alopecic diseases (i.e., alopecia areata and telogen effluvium), and various other cutaneous manifestations have been reported more frequently during the pandemic, both following COVID-19 infection and vaccination (1). However, the association between these dermatological conditions and COVID-19 vaccination has been unclear (2-4), especially when these conditions are not COVID-19-specific and have been known to be triggered by other infections, vaccinations, and various stresses. For alopecia areata, the median (IQR) age of 57 (43-68) years old among participants has placed the participants in the age group with the lowest incidence of alopecia areata (0.1 per 1000 person-year) (5). Treatment with immunosuppressants may also contribute to the lower incidence of immune-mediated vaccine-related adverse reactions. Although the incidence of herpes zoster reactivation post-COVID-19 vaccination is higher than the incidence of the unvaccinated population (0.20% vs 0.11%); this number demonstrated that herpes zoster reactivation following COVID-19 vaccination is extremely rare (6). The higher number of reports is primarily the effect of mass vaccination and less so on the vaccine. The incidence above can explain the absence of herpes zoster reactivation and alopecia in this study, given the number of study participants. We discussed the matter of vaccine-related adverse reactions in the revised manuscript (Line 448-468).

Reference

  1. Nakashima C, Kato M, Otsuka A. Cutaneous manifestations of COVID‐19 and COVID‐19 vaccination. J Dermatol [Internet]. 2023 [cited 2023 Dec 17];50(3):280–9. Available from: https://pubmed.ncbi.nlm.nih.gov/36636825/
  2. Ju HJ, Lee JY, Han JH, Lee JH, Bae JM, Lee S. Risk of autoimmune skin and connective tissue disorders after mRNA-based COVID-19 vaccination. J Am Acad Dermatol [Internet]. 2023 [cited 2023 Dec 17];89(4):685–93. Available from: https://pubmed.ncbi.nlm.nih.gov/37187424/
  3. Martora F, Battista T, Ruggiero A, Scalvenzi M, Villani A, Megna M, et al. The impact of COVID-19 vaccination on inflammatory skin disorders and other cutaneous diseases: A review of the published literature. Viruses [Internet]. 2023 [cited 2023 Dec 17];15(7):1423. Available from: https://pubmed.ncbi.nlm.nih.gov/37515110/
  4. Chen J, Cano-Besquet S, Ghantarchyan H, Neeki MM. The incidence of alopecia areata in a COVID-19-vaccinated population: A single-center review. Cureus [Internet]. 2023 [cited 2023 Dec 17];15(12). Available from: https://www.cureus.com/articles/198463-the-incidence-of-alopecia-areata-in-a-covid-19-vaccinated-population-a-single-center-review#!/
  5. Harries M, Macbeth AE, Holmes S, Chiu WS, Gallardo WR, Nijher M, et al. The epidemiology of alopecia areata: a population‐based cohort study in UK primary care. Br J Dermatol [Internet]. 2022;186(2):257–65. Available from: http://dx.doi.org/10.1111/bjd.20628
  6. Hertel M, Heiland M, Nahles S, von Laffert M, Mura C, Bourne PE, et al. Real‐world evidence from over one million COVID‐19 vaccinations is consistent with reactivation of the varicella‐zoster virus. J Eur Acad Dermatol Venereol [Internet]. 2022;36(8):1342–8. Available from: http://dx.doi.org/10.1111/jdv.18184

- Is muscle pain as AE only on the injection sites or it is generalized?

  • The muscle pain listed in the AE is generalised myalgia. Pain at the injection site is categorised as either immediate or delayed. We have clarified this issue in the revised supplemental document 1 and noted where to find them in the revised manuscript (Line 169)

- Please reference for the sentence “Previous studies have demonstrated a higher rate of reported adverse events following immunization (AEFI) in psoriasis patients than in healthy volunteers and other IMDD patients.”

  • Ans: The reference below was added to the manuscript as requested (Line 76).

Chanprapaph K, Seree-aphinan C, Rattanakaemakorn P, Pomsoong C, Ratanapokasatit Y, Setthaudom C, et al. A real-world prospective cohort study of immunogenicity and reactogenicity of ChAdOx1-S[recombinant] among patients with immune-mediated dermatological diseases. Br J Dermatol [Internet]. 2023 [cited 2023 Dec 17];188(2):268–77. Available from: https://academic.oup.com/bjd/article-abstract/188/2/268/6765233?redirectedFrom=fulltext

Falotico, J.M.; Desai, A.D.; Shah, A.; Ricardo, J.W.; Lipner, S.R. Curbing COVID-19 Vaccine Hesitancy from a Dermatological Standpoint: Analysis of Cutaneous Reactions in the Vaccine Adverse Event Reporting System (VAERS) Database. Am J Clin Dermatol 2022, 23, 729-737, doi:10.1007/s40257-022-00715-x.

Valencia López, M.J.; Meineke, A.; Stephan, B.; Rustenbach, S.J.; Kis, A.; Thaçi, D.; Mrowietz, U.; Reich, K.; Staubach-Renz, P.; von Kiedrowski, R.; et al. SARS-CoV-2 vaccination status and adverse events among patients with psoriasis-Data from the German Registries PsoBest and CoronaBest. J Eur Acad Dermatol Venereol 2023, 37, e831-e833, doi:10.1111/jdv.19039.

- Since the immunogenicity of COVID-19 vaccines is higher among psoriasis patients compared to AIBD patients with a reverse trend showed for the vaccine-related adverse reactions). [Ref 12, A real-world prospective cohort study of 441 immunogenicity and reactogenicity of ChAdOx1-S[recombinant] among patients with immune-mediated dermatological 442 diseases. British Journal of Dermatology 2022, 188, 268-277], I would suggest to add more references and explanation of psoriasis aggravation following COVID19 vaccination in controlled studies as the potential down sides. Other than humoral and cellular immunity in psoriasis, other features of fever, myalgia, malaise will be also important to discuss.

  • Ans: The vaccine-related side effects in psoriasis patients and psoriasis aggravation following COVID-19 vaccination have been added to the discussion of the revised manuscript (Line 448-459).

- Due to higher local effects, should patients with active/unstable psoriasis avoid intradermal vaccination?

  • Ans: Although this is not an absolute contraindication, patients with active/unstable psoriasis should avoid further immunological triggers, including any type of vaccination. Intradermal vaccination represents a physical trauma that may lead to Koebner’s phenomenon in these patients. However, if the benefit of vaccination outweighs the risk, such as in the pandemic, the local side effects are manageable by topical corticosteroid as with any other psoriasis lesions based on our experience treating participants in this study. Systemic flare-ups should be closely monitored and promptly managed in these patients. According to our data, the presence of Koebner’s phenomenon did not always lead to systemic flare-ups. This issue has been discussed in the revised manuscript (Line 456-461).

- Are the injection sites the same, because studies have shown that injections into the same site will result in stronger immune reactions? For systemic effects, some experts have suggested injection to different sites, but for local effect (such as in the treatment of verruca), a repeated injection is suggested. Please comment on this.

  • Ans: Emerging evidence suggests that the COVID-19 vaccine BNT162b2 vaccination in the ipsilateral arm could be immunogenically superior to vaccination in the contralateral arm (1). However, the laboratory surrogate used in this study may not be equipped with the ability to detect this benefit because the current data point to the superiority in the neutralising activity (IC50) of anti-S IgG, median spike-specific CD8 T-cell levels, and CTLA-4 expression on spike-specific CD4 T-cell but not in the anti-S IgG level and avidity (1). Exploring this benefit could be an interesting area of further study. In this study, we required the injections to be on the deltoid area without active skin lesions. If both arms are applicable, most patients would prefer to have the injection on their non-dominant side as the injection site pain, if it occurs, may impede their daily activities post-intervention. In principle, vaccinating immune-mediated dermatological diseases aims to induce adequate SARS-CoV-2-specific immunity and minimise non-specific local and systemic immunological reactions. We believe this principle takes precedence over the induction of a specific part of SARS-CoV-2-specific immunity until there is a benefit and risk assessment of inducing such aspects of SARS-CoV-2-specific immunity in immune-mediated dermatological diseases. We have added this issue in the discussion of the revised manuscript (Line 478-480).

Reference

  1. Ziegler L, Klemis V, Schmidt T, Schneitler S, Baum C, Neumann J, et al. Differences in SARS-CoV-2 specific humoral and cellular immune responses after contralateral and ipsilateral COVID-19 vaccination. EBioMedicine [Internet]. 2023 [cited 2023 Dec 17];95(104743):104743. Available from: https://pubmed.ncbi.nlm.nih.gov/37574375/

- There is imbalance of no systemic immunosuppressants between groups, and the potential impact should be discussed in more depth.

  • Ans: The effect of the systemic immunosuppressant use was accounted for by adjusting during the multivariate analysis. The coefficient of this factor was given to demonstrate its relative effects on the outcomes in the result section and discussed in the revised manuscript (Line 278-280, 298-303).

- Please lists a pro-con comparison. I think it should cover difference in pain, volume, use of adjuvant, speed of mass vaccination, and the disease to be prevented, etc. 

  • Ans: We have added this aspect to the discussion of the revised manuscript (Line 400-416)

- Other than polio, Covid19, and influenza, are there other examples of intradermal and intramuscular studies?

  • Ans: there is also vaccines against rabies, hepatitis B, hepatitis A, measles, DTP, HPV, JE, meningococcal disease, varicella zoster, yellow fever, and vaccinia (1-2). But only intradermal influenza and polio vaccines were tested in immunocompromised hosts. We have added this information in the revised manuscript (Line 417-419).

References

  1. Schnyder JL, De Pijper CA, Garcia Garrido HM, Daams JG, Goorhuis A, Stijnis C, et al. Fractional dose of intradermal compared to intramuscular and subcutaneous vaccination - A systematic review and meta-analysis. Travel Med Infect Dis [Internet]. 2020 [cited 2023 Dec 17];37(101868):101868. Available from: https://pubmed.ncbi.nlm.nih.gov/32898704/
  2. Wilck MB, Seaman MS, Baden LR, Walsh SR, Grandpre LE, Devoy C, et al. Safety and immunogenicity of modified Vaccinia Ankara (ACAM3000): Effect of dose and route of administration. J Infect Dis [Internet]. 2010 [cited 2023 Dec 17];201(9):1361–70. Available from: https://pubmed.ncbi.nlm.nih.gov/20350191/

- In addition to intradermal and intramuscular, what about subcutaneous vaccination?

  • Ans: Subcutaneous vaccination is another alternative route to the intramuscular vaccination alongside intradermal and mucosal routes (e.g., inhaled, oral, eyedrops) (1). The subcutaneous route offers less immediate injection site pain and a lower risk of bleeding and nerve injury but is more suitable for non-adjuvanted vaccines as it is associated with a higher rate of delayed local reactions (2). Despite the absence of added adjuvants, the mRNA and lipid nanoparticle (LNP) components of the BNT162b2 COVID-19 vaccine have inherent adjuvant properties (3); the delayed subcutaneous reaction to the vaccines was reported from a few cases of inadvertent subcutaneous injection (4-5). Like the intradermal route, subcutaneous injection is an off-label indication for the BNT162b2 COVID-19 vaccine, requiring further efficacy and reactogenicity assessment. To our knowledge, data on the immunogenicity of subcutaneous BNT162b2 COVID-19 vaccines has not been limited, with only a small study suggesting a high rate of seroconversion (based on positive anti-S IgG) without an active comparator (6). We have added the discussion of these alternative routes in general and COVID-19-specific context to the revised manuscript (Line 400-416)

References

  1. Zhang L, Wang W, Wang S. Effect of vaccine administration modality on immunogenicity and efficacy. Expert Rev Vaccines [Internet]. 2015 [cited 2023 Dec 17];14(11):1509–23. Available from: http://dx.doi.org/10.1586/14760584.2015.1081067
  2. Cook IF. Subcutaneous vaccine administration – an outmoded practice. Hum Vaccin Immunother [Internet]. 2021 [cited 2023 Dec 17];17(5):1329–41. Available from: https://pubmed.ncbi.nlm.nih.gov/32991241/
  3. Xie C, Yao R, Xia X. The advances of adjuvants in mRNA vaccines. NPJ Vaccines [Internet]. 2023 [cited 2023 Dec 17];8(1):1–6. Available from: https://www.nature.com/articles/s41541-023-00760-5
  4. Ng JY. Inadvertent subcutaneous injection of COVID-19 vaccine. Postgrad Med J [Internet]. 2021 [cited 2023 Dec 17];97(1148):400–400. Available from: https://academic.oup.com/pmj/article/97/1148/400/6969571?login=false
  5. Gyldenløve M, Skov L, Hansen CB, Garred P. Recurrent injection‐site reactions after incorrect subcutaneous administration of a COVID‐19 vaccine. J Eur Acad Dermatol Venereol [Internet]. 2021;35(9). Available from: http://dx.doi.org/10.1111/jdv.17341
  6. Friedensohn L, Zur M, Timofeyev M, Burshtein S, Ben Michael Y, Fink N, et al. Sub-cutaneous Pfizer/BioNTech COVID-19 vaccine administration results in seroconversion among young adults. Vaccine [Internet]. 2021 [cited 2023 Dec 17];39(42):6210–2. Available from: http://dx.doi.org/10.1016/j.vaccine.2021.07.096

Reviewer 3 Report

Comments and Suggestions for Authors

The manuscript represents the results of a clinical trial comparing intradermal and intramuscular application of the SARS-CoV-2 vaccines. The article represents an important approach for clinical dermal autoimmune diseases as well as other dermatological disease. However, the trial has several points that need to be properly described and interpreted. In the primary series, why did the authors used viral vector vaccines and inactivated as the primary source for the first doses? How can equivalent doses be applied intradermally since the viscosity differs depending on the vaccine? Why not use a protein-based vaccine for the third dose? The trial should have a group of control individuals in which intradermal vaccines were administered to determine the effectiveness of the inoculation. In all the studies published with all available SARS-CoV-2 vaccines, even  patients with primary or secondary immunodeficiencies respond to the vaccines when properly applied. In these groups, the mRNA vaccines are the most efficient. It follows then that the authors should have determined the impact of the last vaccine.

There are doubts about the data concerning the cellular immune response. Why the value of 200 mIU/ml of IFNgamma was considered negative when the values should be adjusted by the treatment the patients receive? How do the authors expect that the patients treated with sulfasalazine produce a high amount of IFNgamma, see 10.1046/j.1365-2567.1999.00849.x

There are several other issues which were not properly discussed. How many individuals suffered from the natural infection and had complications or not, and if so which were more prevalent? The titer of antibodies against N protein should have been determined, especially in the patients treated with the inactivated vaccine and those that suffer from the natural infection. Did any individual have neurological symptoms? The question is based on the higher events of neurological symptoms observed in several autoimmune patients.

There are minor, but important details which have to be considered. 

Figure 1 has to be redrawn. The analysis results can not be stated in the same place as the description of the study is diagrammed. The authors could draw another figure for conclusions. Figure 2 should be redone the numbers are difficult to see and the colors should be changed. Figure 3 has to be modified and the whole data, not the mean should be in the graph. 

The discussion should be modified considering the comparison of the different vaccines, and the importance of the different applications. The authors did not mention the possible benefits of the intranasal vaccine. The conclusions should be rewritten.

The authors should include a section with the limitations of the study.

The English language should be revised. There are several sentences difficult to understand.

Comments on the Quality of English Language

The text requires revision.

Author Response

We would like to express our deepest gratitude for your review and constructive comments regarding our manuscript titled “Immunogenicity of intradermal versus intramuscular BNT162b2 COVID-19 booster vaccine in patients with immune-mediated dermatologic diseases: a non-inferiority randomized controlled trial.” submitted to Vaccines. We sincerely appreciate your constructive comments and thank you for given us the opportunity to revise the manuscript. We have made substantial alterations according to your kind suggestion in which we believe have improved our manuscript considerably. Please find our responses your comments below

- In the primary series, why did the authors used viral vector vaccines and inactivated as the primary source for the first doses?

- Why not use a protein-based vaccine for the third dose?

  • Ans. Please allow us to answer these two points collectively. Early in the COVID-19 pandemic, almost everyone living in Thailand received their primary series and the third dose under circumstances where options and supply were limited. People mostly received the vaccines they had access to at the time. Selectively including patients with these uncommon skin diseases who received only specific types of the primary series and third doses was foreseen as an important factor that could lead to an extremely slow recruitment rate. The inclusion criteria, which offer flexibility in participant’s prior vaccination, allow the feasibility of this study during the late COVID-19 pandemic period in Thailand while maintaining the study design that serves the purpose of the study in assessing the performance of intradermal vaccine as a booster dose in this specific patient population. This issue is briefly explained in the methodology section of the revised manuscript (Line 98-100).

- How can equivalent doses be applied intradermally since the viscosity differs depending on the vaccine?

  • Ans. The viscosity of the vaccines given in the investigational and active comparator arms was identical as the vaccine was reconstituted per manufacturer instruction. The dose of the investigational vaccine is a fraction of the dose given via the active comparator arm. The 33% dose fractionation used in this study (10 mg/0.1 ml instead of 30 mcg/0.3 ml) is supported by the immunological equivalent studies in healthy populations and is currently used by the Ministry of Health in Thailand. All prior COVID-19 vaccines received by participants were administered as the standard intramuscular dose of the respective vaccines. Intradermal vaccination was only applied during this study. This issue was clarified in the revised manuscript (Line 114-115)

- The trial should have a group of control individuals in which intradermal vaccines were administered to determine the effectiveness of the inoculation. In all the studies published with all available SARS-CoV-2 vaccines, even patients with primary or secondary immunodeficiencies respond to the vaccines when properly applied.  

  • Ans. We totally agree with you that having a control group to determine the effectiveness of the inoculation is ideal. However, we are concerned that administering another intradermal vaccine for this purpose would be non-reassuring to both the Ethical committee and patients alike, as these patients have heightened cutaneous sensitivity and exhibited the risk of Koebner’s phenomenon. To mitigate the situation, we minimised variation in the inoculation effect by having a single dermatologist perform the injection and photographically document the presence of intradermal wheal for every subject in the interventional arm. This limitation and mitigation strategy was added to the revised manuscript (Line 112-113).

- The mRNA vaccines are the most efficient. It follows then that the authors should have determined the impact of the last vaccine.

  • Ans. Thank you for your valuable suggestion. We reanalysed the data by adjusting the effect of the type of primary series, third dose, and the dose of mRNA vaccines the patient received alongside the other factors in the revised manuscript (Line, Page). With the revised analysis, the non-inferiority conclusions are the same, but we found that the doses of mRNA vaccine received prior to enrolment exert negative effect in the ability to boost-up SARS-CoV-2-specific cellular immunity during this study. We added this information and discussed it in the revised manuscript (Line 298-303, 438-439)

- There are doubts about the data concerning the cellular immune response. Why the value of 200 mIU/ml of IFNgamma was considered negative when the values should be adjusted by the treatment the patients receive?

  • Ans. The value 200 is the test-specific cut-off suggested by the manufacturer based on the validation studies (1-2). Currently, there is no specific cut-off value validated for immunocompromised individuals. In terms of the outcome analysis, the absolute value of IFN-gamma was used as continuous data; hence, the cut-off will not affect the immunogenicity outcome results. This issue is clarified in the revised manuscript (Line 154-159).

References

  1. Saad Albichr I, Mzougui S, Devresse A, Georgery H, Goffin E, Kanaan N, et al. Evaluation of a commercial interferon-γ release assay for the detection of SARS-CoV-2 T-cell response after vaccination. Heliyon [Internet]. 2023 [cited 2023 Dec 16];9(6):e17186. Available from: https://pubmed.ncbi.nlm.nih.gov/37325456/
  2. Huzly D, Panning M, Smely F, Enders M, Komp J, Falcone V, et al. Accuracy and real life performance of a novel interferon-γ release assay for the detection of SARS-CoV2 specific T cell response. J Clin Virol [Internet]. 2022 [cited 2023 Dec 16];148(105098):105098. Available from: https://pubmed.ncbi.nlm.nih.gov/35134681/

- How do the authors expect that the patients treated with sulfasalazine produce a high amount of IFN-gamma, see 10.1046/j.1365-2567.1999.00849.x

  • Ans. Thank you for raising this important matter In this study, we observed that psoriasis patients receiving sulfasalazine can develop a high amount of interferon gamma in the same manner as  patients receiving other medications. Six patients in this study had been prescribed sulfasalazine (1000-3000 mg/day) along with other immunosuppressants and topical steroids years prior to enrolment. Mean IFN-gamma level observed among these participants at 3- and 6-months post-intervention are 4766.5 and 2178.1 mIU/ml respectively. We believed the effect of sulfasalazine on Th1-mediated vaccination outcomes may also depend on the conditions being treated; emphasising the importance of our study which examine vaccination outcomes in the specific patient context.

- How many individuals suffered from the natural infection and had complications or not, and if so which were more prevalent?

  • Ans. Patients with a history of COVID-19 prior to enrolment were excluded from this study (Line 95). The number of patients who had a microbiologically confirmed COVID-19 after enrolment (i.e., breakthrough infection in each arm and the between-group comparison using the Chi-square test is presented in Table 2 under the topic “Participants with breakthrough COVID-19”. There is a slightly but statistically insignificant higher number of participants in the intradermal arm developing breakthrough infection.            

- The titer of antibodies against N protein should have been determined, especially in the patients treated with the inactivated vaccine and those that suffer from the natural infection.

  • Ans. The selection of the immunogenicity serological surrogate in this study aimed to create a homogenised protocol for testing the efficacy of S-protein-based COVID-19 vaccines applicable to participants with mixed prior vaccination backgrounds. Since we excluded patients with prior COVID-19 infection, in our perspective, anti-N protein antibody would be helpful only to 1) explore anti-N antibody dynamics post-intervention in a portion of participants or 2) screen for asymptomatic breakthrough infection occurred during the study (all symptomatic breakthrough infection was tested microbiologically). We are concerned that a test only applicable to some participants could pose significant challenges regarding statistical power and funding justification, so we have omitted the test in this study. However, we fully understand your concern regarding the undetected asymptomatic breakthrough infection and have mentioned this issue in the limitation of the revised manuscript (Line 144-145, 475-478).

- Did any individual have neurological symptoms? The question is based on the higher events of neurological symptoms observed in several autoimmune patients.

  • Ans: We did not observe any abnormal neurological symptoms among participants receiving either arm of the study during the study period. We also received no report concerning study participants developing delayed neurological autoimmune reactions until now. Incidence of various autoimmune diseases increase in association with COVID-19 vaccination during the pandemic (1), all of which we did not observe among our study participants except for the conditions they already had (i.e., psoriasis and autoimmune bullous diseases). The ethnicity of the study participants may explain the difference in the incidence of neurological side effects observed in our and other studies. The most common neurological disorders reported following vaccination are autoimmune demyelinating diseases, which are uncommon in people of Asian descent in the US (2) and less so in Asia Pacific region countries (3,4). Most participants also received systemic immunosuppressants during and after vaccination, which may reduce their risks of developing or manifesting these autoimmune diseases. We discussed the matter of vaccine-related adverse reactions in the revised manuscript (Line 462-471).

References

  1. Peng K, Li X, Yang D, Chan SCW, Zhou J, Wan EYF, et al. Risk of autoimmune diseases following COVID-19 and the potential protective effect from vaccination: a population-based cohort study. EClinicalMedicine [Internet]. 2023 [cited 2023 Dec 16];63(102154):102154. Available from: https://pubmed.ncbi.nlm.nih.gov/37637754/
  2. Hittle M, Culpepper WJ, Langer-Gould A, Marrie RA, Cutter GR, Kaye WE, et al. Population-based estimates for the prevalence of multiple sclerosis in the United States by race, ethnicity, age, sex, and geographic region. JAMA Neurol [Internet]. 2023 [cited 2023 Dec 16];80(7):693. Available from: https://pubmed.ncbi.nlm.nih.gov/37184850/
  3. Cheong WL, Mohan D, Warren N, Reidpath DD. Multiple sclerosis in the Asia pacific region: A systematic review of a neglected neurological disease. Front Neurol [Internet]. 2018;9. Available from: http://dx.doi.org/10.3389/fneur.2018.00432
  4. Tian D-C, Zhang C, Yuan M, Yang X, Gu H, Li Z, et al. Incidence of multiple sclerosis in China: A nationwide hospital-based study. Lancet Reg Health West Pac [Internet]. 2020 [cited 2023 Dec 16];1(100010):100010. Available from: https://pubmed.ncbi.nlm.nih.gov/34327341/

- The discussion should be modified considering the comparison of the different vaccines, and the importance of the different applications. The authors did not mention the possible benefits of the intranasal vaccine.

  • Ans: existing data and corresponding discussion regarding different alternative vaccines (subcutaneous, intranasal routes) in general and specific context (i.e., patients with immune-mediated dermatological diseases) were added to the revised manuscript We have added information on intranasal vaccine as suggested (Line 400-416).

- The conclusions should be rewritten.

  • Ans: The conclusions were revised for your consideration (Line 490-498)

- The authors should include a section with the limitations of the study.

  • Ans: The limitation section is between Line 364-373. In the revised manuscript, we have signposted the section for readers’ convenience at the beginning of the section (Line 473-482).

- The English language should be revised. There are several sentences difficult to understand.

  • Ans: We have tried our very best on correcting grammatical errors. We have also reviewed the manuscript and revised certain complex sentences throughout the manuscript for your consideration.

- Figure 1 has to be redrawn. The analysis results can not be stated in the same place as the description of the study is diagrammed. The authors could draw another figure for conclusions.

  • Ans.The figure was adjusted as suggested.

- Figure 2 should be redone the numbers are difficult to see and the colors should be changed.

  • Ans.The figure was adjusted for more clarity and improvement in quality.

- Figure 3 has to be modified and the whole data, not the mean should be in the graph.

  • Ans.In the revised figure, the red diamond and the black line represent the summary of the whole data as means (the red diamond) and 95%confidence interval (the end of the line pointed at the lower and upper bound). The description of the figure was corrected in the revised manuscript.

Round 2

Reviewer 2 Report

Comments and Suggestions for Authors

Thank you for taking into all the suggestions and made the correction. 

The only minor suggestion is that monkeypox vaccination can also be injected intradermal or intramuscular

Author Response

Thank you for your additional comment. Please find our response below

- The only minor suggestion is that monkeypox vaccination can also be injected intradermal or intramuscular

  • Ans. Thank you for your suggestion. We have listed the study investigating intradermal vs intramuscular moneypox vaccine as one of the references in the revised manuscript (Line 417, Reference 35)

Reviewer 3 Report

Comments and Suggestions for Authors

There was an improvement of the manuscript. There are some issues concerning the equivalence of the vaccine between the two sites of injection. It seems that some of the unwanted effects observed could have been prevented. This should be stated. After the minor amendment the manuscript maz be published.

Comments on the Quality of English Language

Some grammatical mistakes were encountered.

Author Response

Thank you for your additional comments. The revised parts of the manuscript are highlighted yellow for your convenience. Please find our responses to your comments below

- There are some issues concerning the equivalence of the vaccine between the two sites of injection.

  • Ans: We understand your concern about this issue. Emerging evidence suggests that COVID-19 vaccination in the ipsilateral arm could be immunogenically superior to vaccination in the contralateral arm (1). However, the laboratory surrogates used in this study may not be equipped with the ability to detect this benefit because the current data point to the superiority in the neutralising activity (IC50) of anti-S IgG, median spike-specific CD8 T-cell levels, and CTLA-4 expression on spike-specific CD4 T-cell but not in the anti-S IgG level and avidity (1). We believe exploring this benefit of the fID could be an interesting area of further study and have mentioned it in the revised manuscript (Line 480-484).

         Reference

  1. Ziegler L, Klemis V, Schmidt T, Schneitler S, Baum C, Neumann J, et al. Differences in SARS-CoV-2 specific humoral and cellular immune responses after contralateral and ipsilateral COVID-19 vaccination. EBioMedicine [Internet]. 2023 [cited 2023 Dec 17];95(104743):104743. Available from: https://pubmed.ncbi.nlm.nih.gov/37574375/

- It seems that some of the unwanted effects observed could have been prevented. This should be stated.

  • Ans. Thank you for your suggestion. We believe some side effects that may, to a certain extent, be preventable, are the Koebner’s phenomenon. The strategies may include carefully selecting a vaccine inoculation site without active lesions nearby or further optimizing the vaccine’s fractionation. We have stated the issue in the revised manuscript (Line 458-459).